# Autonomous Landing of Quadrotor Unmanned Aerial Vehicles Based on Multi-Level Marker and Linear Active Disturbance Reject Control

**DOI:** 10.3390/s24051645

**Published:** 2024-03-02

**Authors:** Mingming Lv, Bo Fan, Jiwen Fang, Jia Wang

**Affiliations:** 1School of Mechanical Engineering, Jiangsu University of Science and Technology, Zhenjiang 212003, China; 211110201207@stu.just.edu.cn (B.F.); fjw617@just.edu.cn (J.F.); wjjzhb@just.edu.cn (J.W.); 2Fujian Key Laboratory of Green Intelligent Drive and Transmission for Mobile Machinery, Xiamen 361021, China

**Keywords:** unmanned aerial vehicle, autonomous landing, multi-level marker, linear active disturbance reject control

## Abstract

Landing on unmanned surface vehicles (USV) autonomously is a critical task for unmanned aerial vehicles (UAV) due to complex environments. To solve this problem, an autonomous landing method is proposed based on a multi-level marker and linear active disturbance rejection control (LADRC) in this study. A specially designed landing board is placed on the USV, and ArUco codes with different scales are employed. Then, the landing marker is captured and processed by a camera mounted below the UAV body. Using the efficient perspective-n-point method, the position and attitude of the UAV are estimated and further fused by the Kalman filter, which improves the estimation accuracy and stability. On this basis, LADRC is used for UAV landing control, in which an extended state observer with adjustable bandwidth is employed to evaluate disturbance and proportional-derivative control is adopted to eliminate control error. The results of simulations and experiments demonstrate the feasibility and effectiveness of the proposed method, which provides an effective solution for the autonomous recovery of unmanned systems.

## 1. Introduction

In recent years, the collaboration of unmanned aerial vehicles (UAV) and unmanned surface vehicles (USV) has played an important role in water-air cross-domain operations, such as meteorological monitoring, natural exploration, maritime rescue, and so on [1]. The unmanned system consisting of UAVs and USVs can complete river inspection tasks synergistically without interruption, identifying and alerting various abnormal events. However, the UAV always returns to the USV platform for energy charging due to limited battery capacity. To complete tasks like this, the autonomous landing of UAVs is particularly important since this process is a stage with a high failure rate.

Autonomous landing requires comprehensively considering the UAV itself and the USV platform, both of which are not static and suffer complex environmental disturbances [2]. This affects the accuracy and success rate of UAV autonomous landing. Vision-based autonomous landing is a commonly used method, in which one or more specially designed markers are placed on the landing platform. Then, a camera on the UAV captures markers and estimates the relative pose for landing. That is to say, markers are one of the key factors determining autonomous landing [3].

Many markers with different shapes are designed to guide UAV landing, such as H-shaped, T-shaped, QR codes, AprilTags, ArUco, and so on [4]. Furthermore, the scale and number of markers are also important for precise landing, especially in dynamic scenes. It has become a popular trend that multi-level markers are placed on landing platforms for UAV recognition at different heights [5]. This operation supplies more information for pose estimation and autonomous landing.

After planning landing trajectories based on recognition, UAV flight control is another key technology for autonomous landing. Quadrotor UAV is an under-actuated system with six degrees of freedom and four control inputs. Being highly nonlinear, strongly coupled, and complex multi-variable are the characteristics of quadrotor UAVs, which complicate their flight control [6]. Further, it is difficult for UAVs to land precisely according to planned trajectory.

In order to solve the problems described above, a multi-level marker is designed and placed on the USV landing board in this paper, and then the UAV recognizes different markers by a camera and jointly estimates the relative pose. Considering internal and external disturbances of quadrotor UAV, LADRC is employed to control its flight and land autonomously on the USV platform, as shown in Figure 1.

The rest of this paper is organized as follows. In Section 2, the literature related to our work is introduced briefly. In Section 3 and Section 4, we present an autonomous landing method based on joint multi-level identification and LADRC for UAVs. The simulation and experimental results are shown in Section 5. Finally, the conclusions and future work are discussed in Section 6.

## 2. Related Works

The autonomous landing of quadrotor UAVs is a hot topic but not a new issue. There are some good reviews with regard to this aspect, and the landing process requires one or more sensors, such as an inertial measurement unit (IMU), global positioning system (GPS), vision, and so on [7,8].

A QR code-based marker was used to calculate the UAV attitude, and a vision transformer particle region-based convolutional neural network was employed to accelerate feature extraction [9]. The corners of the Apriltag-based marker were detected, and the UAV pose estimation was obtained by rigid body transformation [8]. A marker with three levels was designed and realized for UAV landing by an image-based visual servoing technique [1]. However, it is difficult for all codes at certain levels to be recognized, especially for landing on mobile platforms, such as the USV. An ArUco-based marker was used for scale estimation as well as visual odometry. At the same time, the long-term drift problem was reduced [4].

The landing pad was detected for autonomous landing by AprilTags and color segmentation, the experimental results showed that the UAV landing was successful underground, and the vehicle speed was less than 3 m/s [10]. A hemispherical infrared marker was proposed for the UAV autonomous landing on a moving ground vehicle, and autonomous landing experiments were operated to demonstrate the effectiveness from various angles [11]. An approach based on deep reinforcement learning was designed for the UAV landing on a moving unmanned ground vehicle (UGV), which achieved a high landing success rate and accuracy [12]. This approach did not have any specific communication between the UAV and UGV. However, the premise was to identify the target well.

To solve various interference problems, a landing method based on YOLOv5 and SiamRPN was proposed, in which the proportional-integral-derivative (PID) control was employed as a control law. Then the simulation was carried out on Gazebo 7.16 simulator, in which the effectiveness and robustness were validated [5]. The pan-tilt-based visual servoing system was used for UAV navigating and landing, in which information fusion and signal delay issues had been resolved [13].

Except for the issue of marker and identification, UAV landing control also involves the problem of landing trajectory tracking control, accompanied by various internal and external disturbances. To complete autonomous landing, many control methods have been provided in UAV flight control, for instance, PID control, model prediction control (MPC), neural network control, sliding mode control (SMC), and active disturbance rejection control.

Proportional-derivative (PD) control was employed for UAV flight control, in which the parameters were adapted by particle swarm optimization [14]. The experimental results demonstrated the effectiveness and robustness. A composite control method was proposed for UAV landing, in which disturbances were estimated by an observer and SMC was employed in the feedback channel for landing control [15]. An adaptive robust hierarchical algorithm was proposed to address the impact of rough seas, achieving position tracking of expected trajectories and attitude tracking of command postures [16].

Aiming at eliminating the wave influence on USVs, a bidirectional long short-term memory (BiLSTM) was used to predict its attitude, and PID control was employed to realize UAV-USV synchronous motion. The experimental results show effectiveness in complex marine environments [17]. To solve the problem of height fluctuation in the UAV vertical take-off and landing, active disturbance rejection control (ADRC) was utilized to improve accuracy and rapidity, and the controller parameters were optimized by a multi-strategy pigeon-inspired optimization algorithm [18].

An online nonlinear MPC was proposed for the UAV deep-stall landing in a small space [19]. A model reference adaptive control was employed for parameter adjusting and reduced the UAV landing error [20]. MPC was used for autonomous landing, different from conventional methods, and a synthesized state feedback was formed by the H∞. The method had good performance on disturbance rejection and transient characteristics [21]. In view of the problem of quadrotor UAV flight control being disturbed by wind, a dynamic model based on the wind tunnel test was established, and the response characteristics of discrete and continuous wind disturbances were obtained [22].

In terms of combining vision and control, PID based on a radial basis function and YOLOv3 were employed for UAV landing [23]. To land on a moving vehicle, ApriTag-based makers were designed, and an extended Kalman filter and nonlinear model predictive control were used for UAV autonomous tracking and landing [24]. The PID control was used for UAV landing based on fuzzy logic, in which the parameters of PID were adjusted adaptively [25]. To deal with friction, unmodeled dynamics, and other uncertainties, an adaptive super-twisting control was addressed for UAV vertical take-off and landing [26]. For landing on a UGA, for UAVs, a formation controller was designed [27]. The proposed control structure, which simultaneously considers UGAs and UAVs, was validated by experiment results. The autonomous landing was studied on a UGA for the quadrotor UAV. However, the communication of both was not considered. Instead, a compound AprilTag fiducial marker was employed and a fractional-order fuzzy PID controller was used [28].

In this study, a landing marker based on ArUco is designed in multiple layers for quadrotor UAV autonomous landing on USVs. On the basis of recognition, EPnP (efficient perspective-n-point) is employed to estimate the UAV pose, and the fusion result is obtained by a Kalman filter. Furthermore, UAV landing control adopts LADRC, which mainly includes the disturb estimator and PD control law.

## 3. Landing Marker Detection

Landing marker detection mainly consists of two stages: one is recognizing ArUco and the other is estimating the UAV pose related to the USV.

A landing marker is essential for the UAV to land on the USV platform autonomously. The multi-level marker based on ArUco is employed for pose estimation of the UAV at different heights. As shown in Figure 2, the maximum ArUco is 175 × 175 mm and is placed in the middle of the landing marker, of which the identifier is 19. Four codes with a size of 35 × 35 mm are distributed in four corners, and their identifiers are from 1 to 4, respectively. Then, the smallest ArUco is located in the center of the maximum ArUco, and the identifier is 43.

### 3.1. ArUco Recognition

The ArUco is a composite square marker composed of a wide black border and an internal binary matrix that determines its identifier [4]. One or more ArUco markers may be contained in an image captured by a camera on the UAV. The identifier of each marker is obtained as well as the pixel coordinates of four corner points through image detection, processing, and recognition. The detailed process includes image segmentation, contour extraction and filtration, encoding acquisition and recognition, and corner adjustment.

The purpose of marker recognition is to estimate the UAV pose, which is invaluable for autonomous landing. Owing to its high precision and rapidity, the EPnP (efficient perspective-n-point) is adopted to achieve the pose transformation between the camera and marker coordinate systems. Essentially, it is about finding the rotation matrix and translation vector. The marker coordinate system is defined as the world coordinate system, and then the four corner points of ArUco in the world coordinates are obtained as follows:
(1)$$\begin{array}{l} P_{1}^{w}=[-\frac{l}{2},\frac{l}{2},0{]}^{T},P_{2}^{w}=[\frac{l}{2},\frac{l}{2},0{]}^{T}\\ P_{3}^{w}=[\frac{l}{2},-\frac{l}{2},0{]}^{T},P_{4}^{w}=[-\frac{l}{2},-\frac{l}{2},0{]}^{T} \end{array}$$
where *l* denotes the marker length, and $P_{i}^{w}$(*i* = 1, 2, 3, 4) represents the corner point positions in the world coordinate system.

The EPnP scheme represents the camera coordinates of the reference points as the weighted sum of the control points and then transforms the problem into solving the camera coordinate systems of these four control points. The control points are denoted as $C_{j}^{w}=[x_{j}^{w},y_{j}^{w},z_{j}^{w}]$ and $C_{j}^{c}=[x_{j}^{c},y_{j}^{c},z_{j}^{c}]$ in the world and camera coordinate systems, respectively. The following linear combination can be obtained:(2)$$\left\{\begin{array}{l} P_{i}^{w}=\sum_{j=1}^{4}\alpha_{ij}C_{j}^{w}\\ P_{i}^{c}=\sum_{j=1}^{4}\alpha_{ij}C_{j}^{c}\\ \sum_{j=1}^{4}\alpha_{ij}=1 \end{array}\right.\;P_{i}^{w}$$
where $P_{i}^{c}$ denotes corner point *i* in the camera coordinate system, and $[\alpha_{i1},\alpha_{i2},\alpha_{i3},\alpha_{i4}{]}^{T}$ is the weight vector.

When (*u_i_*, *v_i_*) is the projection of point *i* in the pixel coordinate system, the following equation can be obtained:(3)$$z_{i}[u_{i},v_{i},1{]}^{T}=AP_{i}^{c}=A\sum_{j=1}^{4}\alpha_{ij}[x_{j}^{c},y_{j}^{c},z_{j}^{c}{]}^{T}$$
where *z_i_* is the projection depth, and ***A*** is the internal parameter matrix of the camera, which can be calculated from specific experiments in advance.

Furthermore, the matrix ***A*** contains the pixel focal length (*f_u_*, *f_v_*) and optical center offset (*u_c_*, *v_c_*), and Equation (2) is converted as follows:(4)$$\left\{\begin{array}{c} \sum_{j=1}^{4}\alpha_{ij}f_{u}x_{j}^{c}+\alpha_{ij}(u_{c}-u_{i})z_{j}^{c}=0\\ \sum_{j=1}^{4}\alpha_{ij}f_{v}x_{j}^{c}+\alpha_{ij}(v_{c}-u_{i})z_{j}^{c}=0 \end{array}\right.$$

Eight linear equation systems can be obtained by four pairs of control points and pixel points, and the rotation matrix and translation vector are achieved at last.

### 3.2. Multi-Level Marker Fusion

Although the UAV pose can be generated by a single ArUco, the recognition accuracy is insufficient for autonomous landing because of image noise and so on. When a UAV is at some heights, more than one marker may be recognized, and more accurate pose information can be obtained by the way of fusion.

The landing process can be divided into three stages based on the height between the UAV and USV. In the first stage, there is only the largest ArUco for calculation, and more codes are used in the second stage, of which the number is not fixed. In the last stage, only the smallest ArUco is in the camera’s field of view and is employed to estimate the pose.

A Kalman filter is a method for optimal state estimation for stochastic dynamic systems, of which the state and observer equations are as follows:(5)$$\left\{\begin{array}{l} X_{t}=A_{t,t-1}X_{t-1}+\omega_{t-1}\\ Z_{t}=HX_{t}+v_{t} \end{array}\right.$$
where $X=[p_{x}(t),\;p_{y}(t),\;p_{z}(t),\;\;v_{x}(t),\;\;v_{y}(t),\;\;v_{z}(t)]$ denotes the UAV state, $A_{t,t-1}$ is the state transition matrix, ***H****_t_* is the observer matrix, $\omega_{t-1}$ and *v_t_* are the process noise and the observer noise, respectively, and both are white noise with zero mean. $Z_{t}=[p_{x}(t),\;p_{y}(t),\;p_{z}(t)]$ represents the observer vector.

Assuming the number of identifications for ArUco is *n*, and the estimated locations are [*p_xi_*(*t*) *p_yi_*(*t*) *p_zi_*(*t*)], *i* = 1, 2, …, *n*, then
(6)$$p_{x}(t)=\frac{1}{n}\sum_{i=1}^{n}p_{xi}(t),p_{y}(t)=\frac{1}{n}\sum_{i=1}^{n}p_{yi}(t),p_{z}(t)=\frac{1}{n}\sum_{i=1}^{n}p_{zi}(t)$$

## 4. Landing Control Method

### 4.1. UAV Dynamics

A UAV is composed of a cross bracket and four motors with propellers. Every motor generates a certain amount of torque related to its speed, the combination of which can produce six degrees of freedom for movements of the quadrotor UAV. As shown in Figure 3, two reference frames are introduced to establish the UAV kinematic model. One is the earth-fixed frame ***E***, and the other is the body-fixed frame ***B***.

Let ***P***_e_ and ***V***_e_ be the position and velocity in frame ***E***, respectively; then $V_{e}={\dot{P}}_{e}$. The UAV attitude $\Theta =[\phi ,\theta ,\psi {]}^{T}$ includes the roll, pitch, and yaw angles. The relationship between Θ and rotation speed $\omega_{b}$ in the frame ***B*** is as follows:(7)$$\dot{\Theta }=W\omega_{b}$$
where $W=\left[\begin{array}{ccc} 1 & \tan\theta \sin\phi & \tan\theta \cos\phi \\ 0 & \cos\phi & -\sin\phi \\ 0 & \sin\phi /\cos\theta & \cos\phi /\cos\theta \end{array}\right]$.

According to Newton’s Second Law, the motion equation of the UAV is as follows:(8)$$m{\dot{V}}_{e}=mg-F_{e}+F_{a}$$
where *m* is the UAV’s mass, and ***g*** = [0 0 *g*]^T^ expresses the vector of gravitational acceleration. ***F****_e_* is the resultant force of the UAV, while ***F****_a_* represents the air resistance, which is related to ***V****_e_* and a resistance coefficient.

Assuming the UAV lift force is *f*, and ***V***_e_ = [*v*_x_, *v*_y_, *v*_z_], the following can be obtained:
(9)$$\left\{\begin{array}{l} {\dot{v}}_{x}=-\frac{f}{m}(\cos\psi \sin\theta \cos\phi +\sin\psi \sin\phi )\\ {\dot{v}}_{y}=-\frac{f}{m}(\sin\psi \sin\theta \cos\phi -\cos\psi \sin\phi )\\ {\dot{v}}_{z}=g--\frac{f}{m}\cos\phi \cos\theta \end{array}\right.$$

Based on the Euler equation, the resultant moment of the UAV is as follows:(10)$$J{\dot{\omega }}_{b}+\omega_{b}\times J\omega_{b}=G_{a}+M_{p}$$
where ***J*** denotes the inertia matrix, ***G***_a_ is gyroscopic torque and ***M***_p_ represents the torque generated by the propellers, including the roll, pitch, and yaw.

Let $\omega_{b}$ = [*p*, *q*, *r*] be three components in the frame ***B***. The gyroscopic torque ***G***_a_ is as follows:(11)$$G_{a}=\left[\begin{array}{c} J_{w}q(\Omega_{1}-\Omega_{2}+\Omega_{3}-\Omega_{4})\\ J_{w}p(-\Omega_{1}+\Omega_{2}-\Omega_{3}+\Omega_{4})\\ 0 \end{array}\right]$$
where ***J***_w_ is the total moment of inertia, and Ω*_i_* (*i* = 1, 2, 3, 4) denotes the speed of motor *i*.

Substituting Equation (10) into Equation (11) yields the following:
(12)$$\left\{\begin{array}{l} \dot{p}=\frac{1}{J_{x}}[M_{x}+qr(J_{y}-J_{z})-J_{w}q\Omega ]\\ \dot{q}=\frac{1}{J_{y}}[M_{y}+qr(J_{z}-J_{x})+J_{w}p\Omega ]\\ \dot{r}=\frac{1}{J_{z}}[M_{z}+pq(J_{y}-J_{z})] \end{array}\right.$$
where $J=\left[\begin{array}{ccc} J_{x} & & \\ & J_{y} & \\ & & J_{z} \end{array}\right]$.

The controller of a quadrotor UAV is divided into a position controller and an attitude controller, in which the feedback of the position and attitude can be obtained, respectively, as shown in Figure 4. The inputs of the position controller are the desired position (*x_d_*, *y_d_*, *z_d_*) and rolling angle *φ_d_*, and the outputs, the pitching angle *θ_d_*, yawing angle *φ_d_*, and control signal *u*_1_, are solved. *θ_d_* and *φ_d_* are also the inputs of the attitude controller, which is used to produce *u*_2_, *u*_3_, and *u*_4_.

### 4.2. Linear Active Disturbance Reject Control

ADRC is a novel control method that is an improvement on the traditional PID control. The ADRC components all use nonlinear functions, and many parameters need to be adjusted. Thus, LADRC is proposed and simplifies parameters into an observer bandwidth and a control bandwidth, making the tuning of control parameters simple.

LADRC does not rely on the precise mathematical model of the object. Unknown factors, uncertain states, and external disturbances in the system are considered as the total disturbance of the system, estimated by a linear observer, and compensated by the PD control.

Assuming that the total disturbance includes internal and external disturbances, the dynamic model of a quadrotor UAV is as follows:(13)$$\left\{\begin{array}{l} \ddot{\phi }=b_{\phi }u_{\phi }+f_{\phi }\\ \ddot{\theta }=b_{\theta }u_{\theta }+f_{\theta }\\ \ddot{\psi }=b_{\psi }u_{\psi }+f_{\psi } \end{array}\right.$$

Let ***y*** = [*ϕ*, *θ*, *ψ*], ***x***_1_ = ***y***, ***x***_2_ = $\dot{y}$, ***f*** = [*f_ϕ_*, *f_θ_*, *f_ψ_*], and ***x*** = [***x***_1_, ***x***_2_, ***f***]^T^, and the extended state space equation of the UAV is described as follows:
(14)$$\left\{\begin{array}{l} \dot{x}=Mx+Nu+E\dot{f}\\ y=Ox \end{array}\right.$$
where the state matrix $M=\left[\begin{array}{ccc} 0 & 1 & 0\\ 0 & 0 & 1\\ 0 & 0 & 0 \end{array}\right]$, the input matrix $N=\left[\begin{array}{ccc} 0 & 0 & 0\\ b_{\phi } & b_{\theta } & b_{\psi }\\ 0 & 0 & 0 \end{array}\right]$, the control matrix $u=\left[\begin{array}{ccc} 0 & 0 & 0\\ u_{\phi } & u_{\theta } & u_{\psi }\\ 0 & 0 & 0 \end{array}\right]$, the disturbance matrix $E=\left[\begin{array}{c} 0\\ 0\\ 1 \end{array}\right]$, and the output matrix $O=\left[\begin{array}{ccc} 1 & 0 & 0 \end{array}\right]$.

LESO is the key to achieving active disturbance rejection control. When designing LESO, it is necessary to select an appropriate feedback gain matrix to ensure that the observation error converges to zero. In addition, the dynamic response of the observer is considered to ensure the accuracy and reliability of the observation results. According to LADC, the LESO of Equation (14) is as follows:
(15)$$\left\{\begin{array}{l} \dot{z}=Mz+Nu+L(y-\hat{y})\\ \hat{y}=Oz \end{array}\right.$$
where ***z*** is the observed value of ***x***, $\hat{y}$ denotes the estimated output, including ***z***_1_ and ***z***_2_, which correspond to ***x***_1_ and ***x***_2_, and ***L*** is the gain matrix of the observer error feedback. In [29], the poles of the characteristic equation are put in the same place, then $L=[3\omega_{0},3\omega_{0}^{2},\omega_{0}^{3}]$, wherein *ω*_0_ represents the observer bandwidth of LESO.

In LADRC, the control law employs the PD control, that is
(16)$$u_{0}=k_{p}(r-z_{1})-k_{d}z_{2}$$
where ***u***_0_ is the control quantity, ***r*** is the desired signals, including the roll, pitch, and yaw angles, and ***k***_p_ and ***k***_d_ are the proportional and differential gains of PD control, respectively.

## 5. Simulation and Experiment

### 5.1. Platform

As shown in Figure 5, the basic frame of this P450-Nano UAV is composed of composite materials, reducing the overall weight to 1950 g with the battery (4000 mAh). The size is 335 × 335 × 230 mm, while the wheelbase is 410 mm, and the maximum payload is 1600 g. Four brushless motors are distributed, and the model is T-motor-2216. An onboard camera with a 1920 × 1080 maximum resolution and a 3.6 mm focal length is mounted below the UAV body to acquire the landmark images. The online computer with a Cortex-A57 CPU and a 128 NVIDIA Maxwell GPU is employed to control flights and process images. Our USV is made of recyclable ABS engineering plastics. The maximum speed is 2 m/s, and the total weight is 8 kg including two batteries. The software is developed on ROS-melodic (Robot Operating System) based on ubuntu18.04. The proposed multi-level ArUco markers are placed on a landing board, of which the length and width are both 0.5 m. In LADRC, the observer bandwidth *ω*_0_ is set as 50, the proportional gain *k_p_* is 0.45, and the differential gain *k_d_* is 0.17.

### 5.2. Simulation

A simulation model of UAV autonomous landing is built on Gazebo, which provides physical simulations with high fidelity and a user-friendly interaction mode. Our simulation is to verify the feasibility of vision-based landing, including searching, adjusting, and landing stages as shown in Figure 6.

The UAV takes off from a starting point of zero, climbs to a 1 m height at a fixed speed, and then activates the searching command. When position information recognized by the UAV meets the landing condition threshold, the adjusting and landing tasks are executed. If the UAV cannot recognize the landing marker, it slowly rises to search again.

When the landing marker is recognized successfully, the current position of the UAV is used as a starting point to expand the cruising range in a clockwise direction, with a square trajectory. In this process, the UAV attitude is adjusted for the final landing. Figure 6 illustrates the effectiveness of the proposed landing markers.

### 5.3. Ground Experiment

An experiment is carried out by placing the proposed multi-level ArUco markers on the ground. The UAV starts landing autonomously when the flight height is 10 m. As shown in Figure 7, the largest ArUco marker (ID = 19) is recognized at 7.7 m, while the smallest ArUco marker (ID = 3) is at a height of about 0.8 m. In this process, at least one medium-sized marker can be acquired, but the quantity is uncertain, which shows that the multi-level marker is effective and necessary.

Furthermore, the multi-layer recognition method is discontinued, only identifying the single marker in the middle of the landing board. The ground experiments are conducted 30 times, and the landing error of the two methods is shown in Figure 8. The root mean square errors are 0.082 m and 0.035 m, respectively. Obviously, the multi-level marker method achieves better landing accuracy.

### 5.4. Surface Experiment

Another experiment is conducted by placing the proposed multi-level markers on a USV, which was developed in a previous project. This experiment is an autonomous landing without GPS assistance. The initial height of the UAV is about 1.3 m, and the maximum height recognized is about 9 m. When approaching 0.2 m, the motors are turned off, and the UAV lands on the landing board freely. The process is shown in Figure 9, in which the left part of each picture is the image captured by the onboard camera, and the right part is from a specialized recording camera. In the landing process, there is more disruption than landing on the ground.

To demonstrate the accuracy of the proposed landing method, a comparison is made by PID control, in which the proportional gain, integral gain, and differential gain are set as 0.45, 0.052, and 0.17, respectively. It should be noted that the two methods use the same markers as the proposed ones. The trajectories of the UAV landing and the USV motion are shown in Figure 10. Both methods can land successfully, but it is observed that the landing curve of LADRC is smoother. The landing accuracy of LADRC is 0.057 m, while PID is 0.11 m. The results indicate that LADRC may effectively resist internal and external disturbances.

From the results of the ground and surface experiments, it can be seen that flying too high or too low may cause incomplete marker information problems, which may affect UAV autonomous landing on the USV platform. The proposed multi-level markers can effectively solve this problem. At the same time, to follow the expected landing trajectory, the flight control method based on LADRC has excellent landing accuracy.

## 6. Conclusions and Future Work

In this study, a method of UAV landing on USVs autonomously was investigated. A multi-level landing marker was placed on the USV landing board and captured by a camera mounted below the UAV body. The land marker based on ArUco contained three levels and five codes and was employed for UAV landing at different heights. Then, the EPnP algorithm was adopted to achieve the UAV pose and fusion results by different ArUco improved landing accuracy and stability. Further, LADRC based on the controller was designed for landing control. The disturbances were estimated by a linear extended state observer and eliminated by PD control. Then, the UAV could land on the USV autonomously with high precision, which was demonstrated by simulations and experiments. The proposed method enables the UAV to land stably on the USV platform for charging and expanding motion purposes.

In upcoming work, achieving UAV landing on USVs with complex motion will be the focus of research, and how to compensate for wave impacts on USV motions will be another research topic.

## Figures and Tables

**Figure 1 sensors-24-01645-f001:**
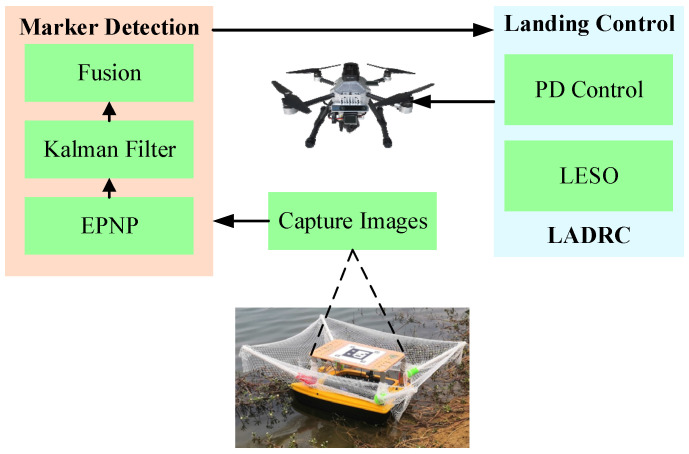
The structure of the proposed method. The method includes marker detection and landing flight control, the purpose of which is to land on the USV platform autonomously for the UAV.

**Figure 2 sensors-24-01645-f002:**
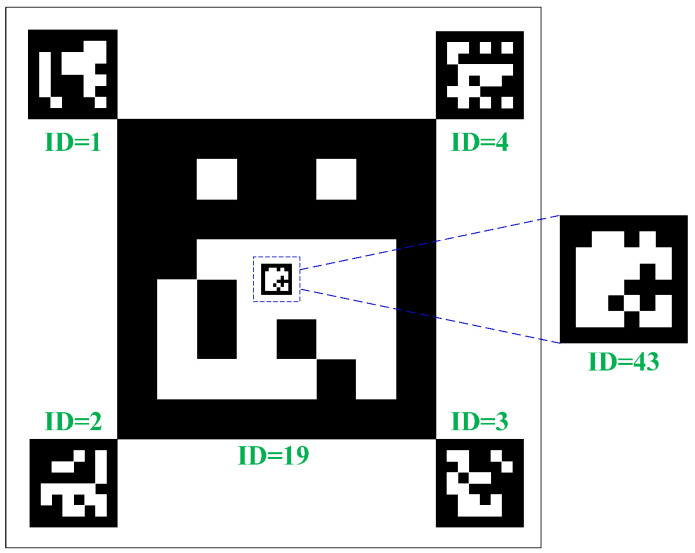
The distribution of the proposed multi-level landing marker. The first level is the ArUco of which the ID is 19, and the second level contains four ArUcos of which the ID are 1, 2, 3, and 4, respectively, while the third level is the ArUco and the ID is 43.

**Figure 3 sensors-24-01645-f003:**
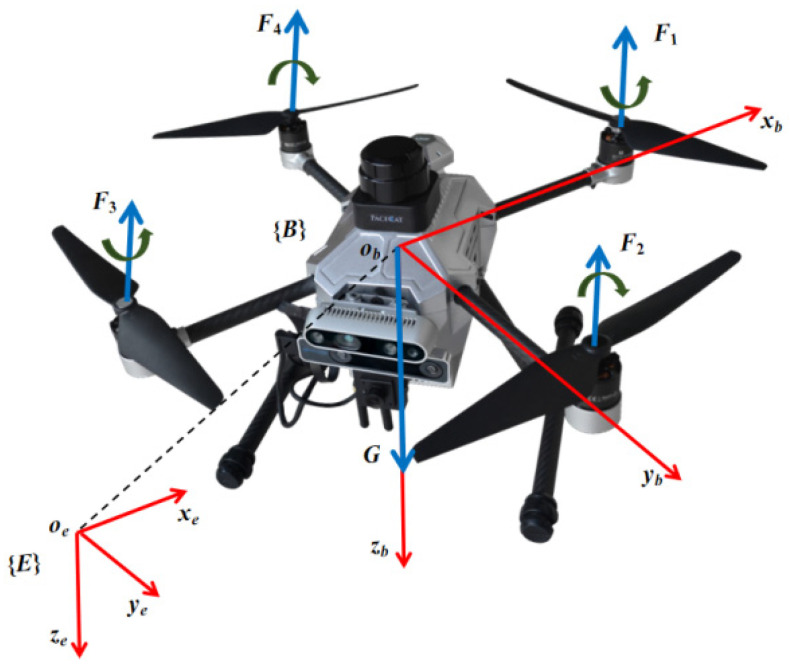
The quadrotor UAV reference frames. The red arrows represent coordinate frameworks, and blue arrows denotes force directions. {***E***} represents earth-fixed reference frame, while {***B***} denotes body-fixed reference frame. ***F***_1_ to ***F***_4_ are thrusts generated by four motors, respectively, and G represents the gravity of the UAV.

**Figure 4 sensors-24-01645-f004:**
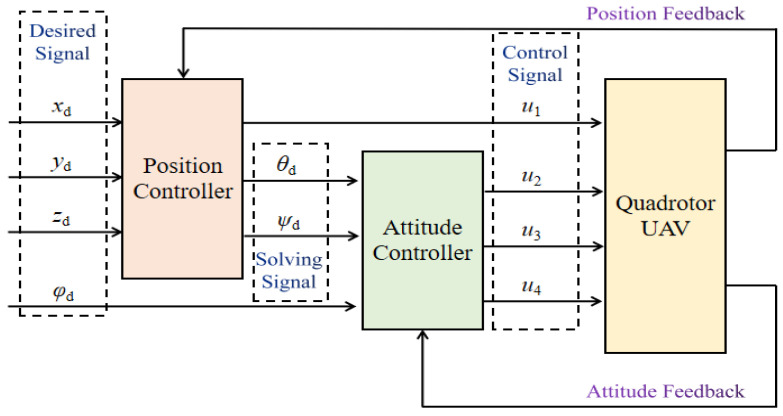
Control schematic of UAV landing. The controller includes a position controller and an attitude controller.

**Figure 5 sensors-24-01645-f005:**
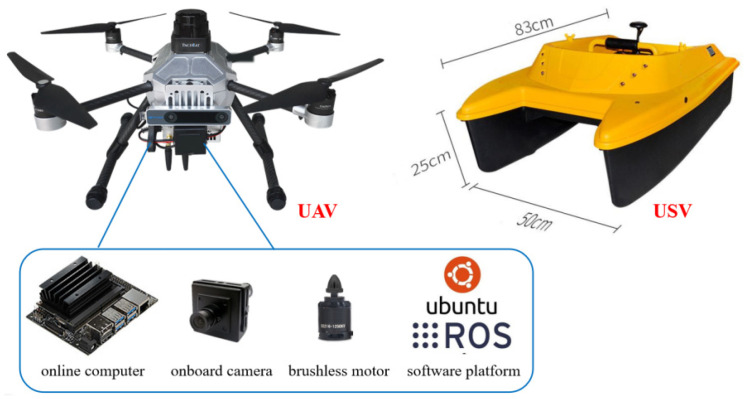
The picture of our UAV and USV with the main components. The hardware includes a body frame, an online computer, an onboard camera, and four brushless motors, while the software is based on Ubuntu 18.04 and ROS-melodic.

**Figure 6 sensors-24-01645-f006:**
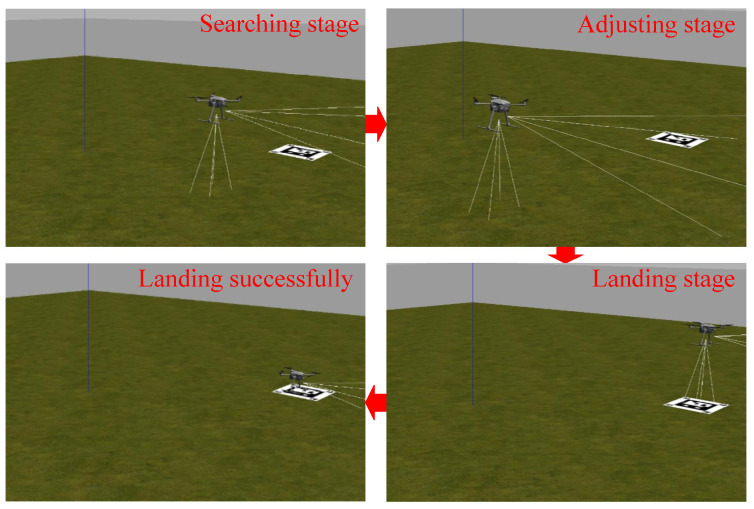
The simulation result. Three main stages are shown in the simulation, including searching, adjusting, and landing. Finally, the UAV lands on the ArUco markers successfully.

**Figure 7 sensors-24-01645-f007:**
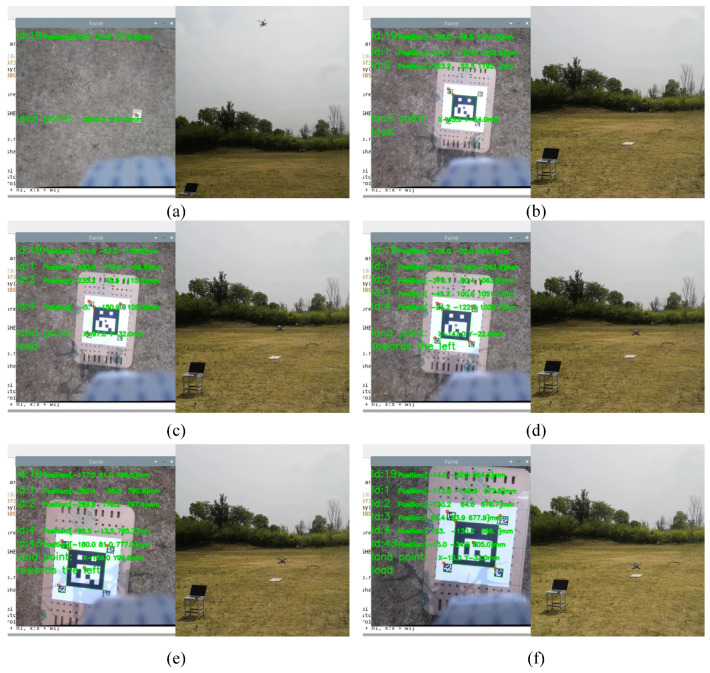
The ground experiment results; (**a**) recognizes the largest marker, and its ID is 19, (**b**) is 19, 1, and 2, (**c**) is 19, 1, 2, and 4, (**d**) is 19, 1, 2, 3, and 4, (**e**) is 19, 1, 2, 4, and 43, while (**f**) recognizes all markers, respectively.

**Figure 8 sensors-24-01645-f008:**
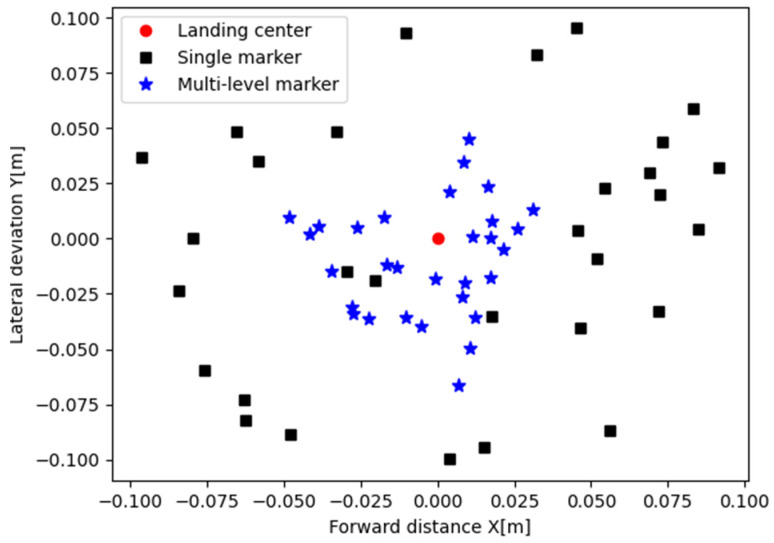
The ground landing errors from 30 ground experiments. From the results, it can be seen that the multi-level marker performs better than the single marker.

**Figure 9 sensors-24-01645-f009:**
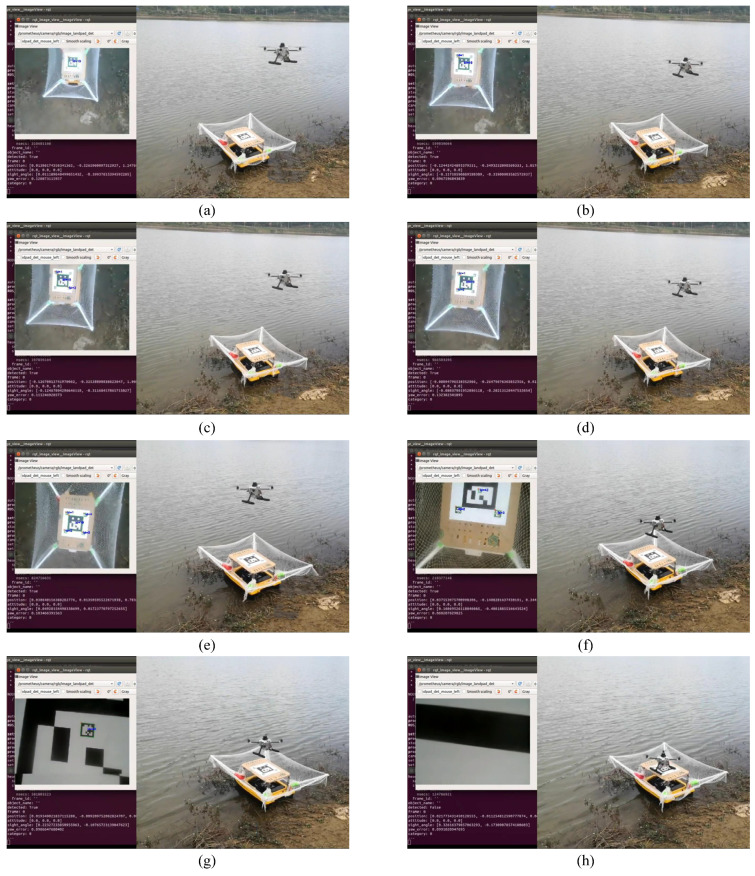
The surface experiment results. (**a**–**h**) show the process of UAV autonomous landing, in which different markers are recognized for adjusting the UAV pose.

**Figure 10 sensors-24-01645-f010:**
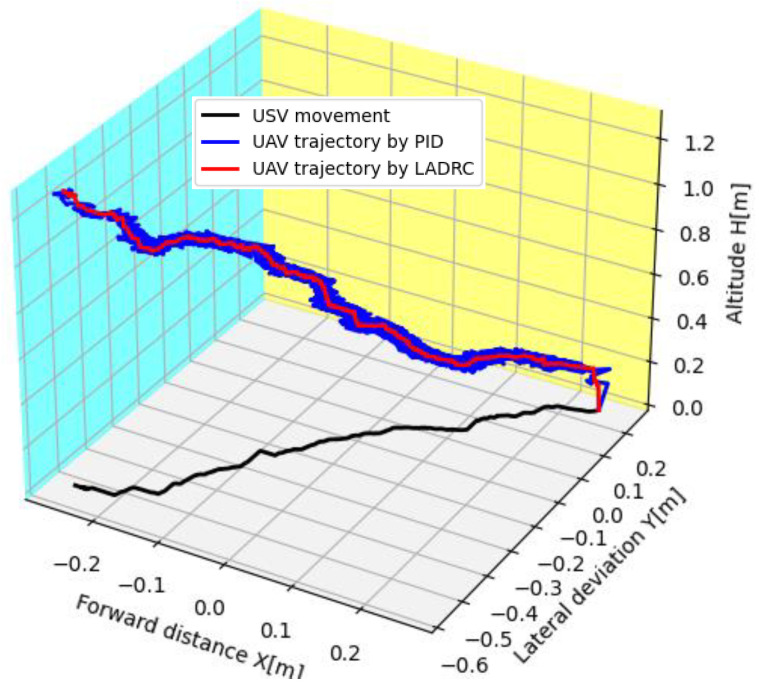
The landing curves. Two control methods are used for comparison: one is PID control and the other is LADRC, which was employed in our work.

## Data Availability

The data are unavailable due to privacy restrictions.

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
