# Peer review of "Autonomous Landing of Quadrotor Unmanned Aerial Vehicles Based on Multi-Level Marker and Linear Active Disturbance Reject Control"

_sensors, 2024, doi:10.3390/s24051645_

Round 1

Reviewer 1 Report

Comments and Suggestions for Authors

This paper discusses an autonomous landing method for UAVs on USVs. It introduces a multi-level marker system for enhanced landing accuracy and employs a LADRC technique to handle disturbances and control errors. The approach is validated through simulations and field experiments, demonstrating the effectiveness and robustness of the proposed approach. Some issues can be fixed to improve the overall clarity of the paper.

 Minor Issues 

1. The definition of LADRC is ambiguous, I suggest include a detailed explanation of the LADRC technique, including its specific parameters and deployment.

 2. It would be valuable to include a comparative analysis with other UAV landing techniques on ground like the work of [1-2] to highlight the advantages and potential limitations of the proposed method, e.g.,

Wang, Chang, et al. "Vision-Based Deep Reinforcement Learning of UAV-UGV Collaborative Landing Policy Using Automatic Curriculum." Drones 7.11 (2023): 676. 

3. More information regarding environmental conditions and UAV specifications should be included in experimental setup to provide better context for the results. 

4. Data Analysis: A deeper analysis of the landing accuracy data would strengthen the results section, possibly including statistical significance tests or error distribution analysis.

Comments on the Quality of English Language

There are many errors in grammar, e.g., "The results of simulation and experiments." Should be “simulations”, 'H shape' and 'T shape' should be 'H-shaped' and 'T-shaped', 'kalman filter' should be 'Kalman filter', "In recent years collaborative of" where 'collaborative' should be 'collaboration'. Please check and polish the paper.

Reviewer 2 Report

Comments and Suggestions for Authors

The authors investigated the Autonomous Landing of Quadrotor UAV and proposed the UAV Based on Multi-level Marker and Line-ar Active Disturbance Reject Control method. It's an intriguing study; however, the current version of the manuscript requires further modifications. My recommendations for minor revisions are as follows:

(1) Attention should be given to formatting requirements. For instance, '[19] proposed a ...' is not in line with standard citation formatting.

(2) The experiments seem insufficient. The authors should consider comparing their proposed algorithm with the latest algorithms available.

(3) There is limited analysis of the experimental results. The authors should provide a detailed analysis of the characteristics of the proposed algorithm—what makes it superior and its specific strengths.

(4) The experimental content is limited. I suggest the authors conduct additional experiments and perform statistical analyses on multiple sets of data.

Comments on the Quality of English Language

nothing

Reviewer 3 Report

Comments and Suggestions for Authors

This research claim developing autonomous landing method ased on a multi-level marker and linear active disturbance rejection 10 control (LADRC). A specially designed landing board was placed on USV and ArUco codes with different scales are employed. Then, the landing marker was captured and processed by a camera mounted below the UAV body. Using efficient perspective-n-point method, position and attitudes of UAV are estimated and further fused by kalman filter, which improves estimation ac-14 curacy and stability. On this basis, LADRC was used for UAV landing control, in which extended 15 state observers with adjustable bandwidth is employed to evaluate disturbance and proportional derivative control was adopted to eliminate control error. There are some comments that must be addressed:

The authors are advised to provide a more compelling introduction that clearly highlights the significance of the problem addressed and its relevance in the context of unmanned systems.

To make this paper better, authors need to improve their opening so as to give a clear account on why their study pertains to their problem areas within unmanned systems.

In section 3, the authors are advised to elaborate further on the multi-level marker and linear active disturbance rejection control (LADRC) methodology.

The authors are advised to include more details about the experimental setup, such as the specifications of the landing board, characteristics of the ArUco codes used, and any specific conditions considered during simulations and experiments.

The authors are advised to consider adding a sentence or two on potential future directions or applications of their proposed autonomous landing method. This could provide a broader context for the significance of their work.

Remember, these comments are meant to guide the authors in improving specific aspects of their abstract. Providing constructive feedback will help them refine their work.

Ensure that all figures are clear and easily interpretable (check Figure 2). The authors are advised to use high-resolution images and graphics to enhance visibility. Also, the captions should adequately explain the content and purpose of the figures.

The authors are advised to ensure that tables are easy to read and understand. Use concise headings, and avoid unnecessary details that may clutter the tables.

In sections 3, 4, The authors are advised to ensure that all symbols used in the paper are defined either in the text or in a dedicated symbols/notation section. It‘s also necessary to maintain consistency in the use of symbols throughout the Equations (2-16).

Comments on the Quality of English Language

Extensive editing of English language required

Round 2

Reviewer 1 Report

Comments and Suggestions for Authors

This version has answered my previous concerns. 

Reviewer 3 Report

Comments and Suggestions for Authors

The paper demonstrates excellent organization and clarity in its presentation, and the authors have adeptly addressed all previously raised points and inquiries.

Comments on the Quality of English Language

Minor editing of English language required